# Anti-Herpes Simplex 1 Activity of *Simmondsia chinensis* (Jojoba) Wax

**DOI:** 10.3390/molecules26196059

**Published:** 2021-10-07

**Authors:** Zipora Tietel, Sarit Melamed, Noy Eretz-Kdosha, Ami Guetta, Raanan Gvirtz, Navit Ogen-Shtern, Arnon Dag, Guy Cohen

**Affiliations:** 1Gilat Research Center, Department of Food Science, Agricultural Research Organization, Mobile Post Negev 8531100, Israel; saritm@volcani.agri.gov.il; 2The Dead-Sea and Arava Science Center, The Skin Research Institute, Masada 86910, Israel; Noy@adssc.org (N.E.-K.); ami.guetta2000@gmail.com (A.G.); Raanan@adssc.org (R.G.); Navit@adssc.org (N.O.-S.); 3Eilat Campus, Ben-Gurion University of the Negev, Eilat 8855630, Israel; 4Gilat Research Center, Department of Fruit Tree Sciences, Agricultural Research Organization, Mobile Post Negev 8531100, Israel; arnondag@volcani.agri.gov.il

**Keywords:** Jojoba, medicinal properties, dermatology, bioactive molecules, *Simmondsia chinensis*, simmondsin, virus, HSV-1, Herpes simplex

## Abstract

Jojoba (*Simmondsia chinensis* (Link) Schneider) wax is used for various dermatological and pharmaceutical applications. Several reports have previously shown beneficial properties of Jojoba wax and extracts, including antimicrobial activity. The current research aimed to elucidate the impact of Jojoba wax on skin residential bacterial (*Staphylococcus aureus* and *Staphylococcus epidermidis*), fungal (*Malassezia furfur*), and virus infection (herpes simplex 1; HSV-1). First, the capacity of four commercial wax preparations to attenuate their growth was evaluated. The results suggest that the growth of *Staphylococcus aureus*, *Staphylococcus epidermidis*, and *Malassezia furfur* was unaffected by Jojoba in pharmacologically relevant concentrations. However, the wax significantly attenuated HSV-1 plaque formation. Next, a complete dose–response analysis of four different Jojoba varieties (Benzioni, Shiloah, Hatzerim, and Sheva) revealed a similar anti-viral effect with high potency (EC_50_ of 0.96 ± 0.4 µg/mL) that blocked HSV-1 plaque formation. The antiviral activity of the wax was also confirmed by real-time PCR, as well as viral protein expression by immunohistochemical staining. Chemical characterization of the fatty acid and fatty alcohol composition was performed, showing high similarity between the wax of the investigated varieties. Lastly, our results demonstrate that the observed effects are independent of simmondsin, repeatedly associated with the medicinal impact of Jojoba wax, and that Jojoba wax presence is required to gain protection against HSV-1 infection. Collectively, our results support the use of Jojoba wax against HSV-1 skin infections.

## 1. Introduction

Jojoba (*Simmondsia chinensis* (Link) Schneider) is a Buxaceae dioecious evergreen shrub native to the Sonora and Mojave deserts of Arizona, California and Mexico [1]. At present, Jojoba is cultivated in semi-arid areas worldwide, including in the USA, Israel, India, Australia, Mexico, Chile, Argentina, Peru, and Egypt. Israel is one of the main growers with about 2400 ha. Ref. [2], and high annual average yields of more than 5000 kg of seeds per ha [3].

Jojoba seeds contain 45–55% liquid wax, which can be extracted by cold press. The unique molecular structure of the wax consists of C36–C46 esters, formed mainly by C16-C24 saturated and monounsaturated fatty acids and fatty alcohols [4,5,6]. In addition, the wax also contains tocopherols and phytosterols [7]. The wax was initially used as a substitute for whale sperm, which was banned in the 1970s. Since then, various uses for the wax have been suggested, from biodiesel to industrial uses as a lubricant, thanks to its chemical consistency [8]. The increase in wax production in recent years has made Jojoba more feasible as an industrial crop [9]. Due to the chemical resemblance of the wax to the human glandular sebum, which consists of 2–30% wax esters [10], it was suggested as suitable for cosmetic uses [11]. At present, Jojoba wax is widely used as a valuable raw material in pharmaceutical and cosmetic products.

Jojoba is traditionally known for its skin-related bioactivities. However, there are not many peer-reviewed scientific papers to support this notion. Native Americans used the crushed seed wax for skincare—e.g., treating cuts, scars, bruises, and burns, including sun and windburn, or for hair lubrication [12]. At present, the two main dermatological uses are topical application of the wax as-is, and using it as an ingredient in topical formulations thanks to its favorable physio-chemical properties. Topical applications of Jojoba wax are diverse and include massage therapy, anti-scars and anti-burns treatments, anti-stretch marks, as well as nourishing and treating tattooed skin [13]. In addition, massage professionals, aromatherapists, and aestheticians use it for herbal and natural skincare preparations. Jojoba wax has low topical permeation, forming an efficient barrier protecting the skin surface [14,15]. When applied externally, Jojoba wax leads to a semi-occlusion of the skin surface, maintaining moisture in the skin and reducing transepidermal water loss (TEWL) [14]. The moisturizing and barrier-repair effects might be attributed to ceramides [16], although the lipidomic profile of Jojoba has not been published to date.

Other important reported bioactivities include a remarkable anti-inflammatory effect via a significant decrease in Prostaglandin E_2_ (PGE_2_) content in exudates, while preventing Tumor necrosis factor α (TNF-α) formation [17]. Furthermore, Jojoba wax improved acne in a clay face mask formulation [18] and was proven effective in treating acne and psoriasis [19]. 

The skin is the largest organ of the human body, forming an efficient barrier against the deleterious action of the environment, also serving as a sensory and immune-active organ, and blocking unregulated water and solutes loss (reviewed by [20]). It protects against biological (microbes) and chemical agents (corrosive, irritating, and allergenic substances), and physical factors (radiation, mechanical and thermal factors). In recent years, the importance of the commensal microbiome and mycobiome communities has gained increasing interest. This milieu of microorganisms plays a crucial role in immune training, thus also showing dysbiosis in a variety of disease states—e.g., atopic dermatitis and acne [21]. Metagenomic skin microbiome studies have demonstrated that these populations are site-specific but are mostly stable over time [21,22], with approx. 80% constant populations [23].

*Staphylococcus epidermidis* and *Staphylococcus aureus* are two of the most common human cutaneous bacteria. *S. epidermidis* has been isolated from all skin microenvironments, including dry, moist, sebaceous, and foot regions, with different strains characteristic to each [23]. Interestingly, it has been shown that skin colonization by specific strains of *S. epidermidis* may either aid or hamper the skin barrier [23]. *S. epidermidis’s* more pathogenic cousin, *S. aureus*, is a commensal bacterium of the human nasopharynx and skin, and also a common human pathogen, which can cause a variety of infectious diseases, from skin and soft tissue infections, through endocarditis, osteomyelitis, bacteremia, to lethal pneumonia [24]. *S. aureus* easily acquires antibiotic resistance, and even antibiotic-susceptible cells can survive and develop tolerance to high antibiotic concentrations and are associated with chronic and recurrent infections [25]. *Malassezia furfur* is a lipophilic yeast, part of the normal cutaneous flora in humans. *M. furfur* colonization on skin surfaces has been associated with various dermatological conditions, e.g., atopic dermatitis and dandruff [26,27], while also involved in systemic infections in immunosuppressed patients and neonates [27]. The emergence of resistant fungal strains raises the necessity of searching for natural antifungals sources [28].

*Herpes simplex type 1* (HSV-1) is a widely distributed neurotropic human pathogen—a member of the Alphaherpesviridae subfamily. It is transmitted by intimate contact with an infected individual. After primary infection of epithelial cells, the virus becomes latent in neurons of the peripheral and central nervous systems [29]. It can then be periodically reactivated, resulting in recurrent clinical or subclinical episodes of usually labial infections throughout life [30]. Severe acute infection due to repeated viral reactivations is accompanied by pain, itching, and inconvenience, sometimes in addition to flu-like symptoms. 

The aim of the current work was to evaluate the impact of Jojoba wax on skin-related microbes and HSV-1, in addition to chemically characterizing its fatty acid and fatty alcohol composition. 

## 2. Results and Discussion

The study was initiated by screening the antibacterial (*Staphylococcus epidermidis and Staphylococcus aureus*), anti-fungal (*Malassezia furfur*), and antiviral (*Herpes simplex type 1;* HSV-1) bioactivities of Jojoba wax obtained from commercial industrial lines. These model organisms were selected due to their importance in the skin resident flora and their high prevalence as causes of skin infections [21,23,27]. Minimum Inhibitory Concentration (MIC) assays performed in *S. epidermidis* and *S. aureus* cultures show no significant inhibition of bacterial growth upon treatment with Jojoba wax when added at concentrations of up to 850 µg/mL (Figure 1A,B). Since a kinetic analysis was performed, both the bacteriostatic and bactericidal properties of the wax in these experimental settings were excluded. In contrast, ampicillin (10 µg/mL), used as a positive control group, attenuated the growth of both bacteria. Al-Ghamdi and colleagues have recently shown that Jojoba wax can attenuate the growth of *Bacillus subtilis*, *S. aureus*, *Proteus vulgaris*, and *Proteus mirabilis* [31]. The discrepancy may be explained by the different Jojoba cultivar and culture conditions that can alter the chemical composition of the wax and its anti-microbial capacity. The absence of antimicrobial effect on methicillin-resistant *S. aureus* up to 100 mg/mL using the disc diffusion method was also reported, confirming our results [32]. 

Next, the ability of Jojoba wax to attenuate *Malassezia furfur* growth was investigated in the AlamarBlue assay. Similarly, lack of effect was seen when the wax was supplemented at up to 850 µg/mL (Figure 1C). Negligible anti-fungal activity of Jojoba wax was reported previously in two other pathogenic fungi species (*Candida albicans* and *Asperigillus flavus* [31], suggesting that Jojoba wax is not a potent fungicide agent. At the same time, in some of its applications, e.g., as a lubricant in spa therapy, Jojoba can be topically applied as-is on the skin, allowing exposure to a high concentration of its bioactive ingredients. Thus, it cannot be excluded that the low potency reported by others may still reflect clinical importance [21,23].

Next, the ability of Jojoba wax to attenuate the formation of HSV-1 plaques in Vero cells (host cells) was investigated. Vero cell viability was concomitantly determined by an MTT assay to exclude direct cytotoxicity of the wax. While Jojoba wax at 850 µg/mL was well tolerated by the host cells (Figure 1D), the same concentration of four Jojoba wax preparations significantly inhibited the formation of HSV-1 plaque (approx. 35–55%, Figure 1E). Of note, Yarmolinsky et al. had shown in 2010 the antiviral impact of ethanolic and aquatic leaf extracts of Jojoba. However, Jojoba wax has not been investigated in their research [33]. In addition, patents submitted in 2003 [34] and 2004 [35] describe the use of Jojoba wax or fractions of Jojoba esters for the treatment of herpes lesions. Both patents described the favorable clinical observation in the Jojoba-treated groups [34,35].

To further investigate the antiviral impact of Jojoba, the wax was supplemented within the culture media with low levels of Triton X-100 (see materials and methods) to increase its proper dispersity and bioavailability [36]. The results show that in these settings, Jojoba wax at the highest concentration (1000 µg/mL) was toxic to the cells (Figure 2A). Therefore, a lower concentration range was used in further experiments. Next, the impacts of different Jojoba wax, originating in four cultivars—Benzioni, Shiloah, Hatzerim, and Sheva—were examined. These represent the most common cultivars used in Israel (Jojoba Desert and Jojoba Valley, personal communication). The results in Figure 2B show high potency (EC_50_ 0.96 ± 0.4 µg/mL) and efficacy, resulting in the blockage of HSV-1 plaque formation in vitro in all four Jojoba cultivars. The increased activity of Jojoba wax observed here in comparison to the results shown in Figure 1E suggests that the main limitation in using Jojoba wax against HSV-1 is its reduced bioavailability in the culture media. However, as Triton X-100 has detergent action and HSV-1 is an enveloped virus, the improvement in activity may also be attributed to the synergistic action of Jojoba wax with the surfactant. Further experiments are required to validate this hypothesis. The similar antiviral action of all four Jojoba cultivars and the enhanced effectiveness of the wax upon the supplementation of the surfactant support the involvement of a highly abundant and lipophilic compound. The results in Table 1 show the chemical composition of the different wax preparations. 

The fatty acid composition of Jojoba wax was reported in various works. Fatty acids range from C16:0 to C24:1, with C20:1 as the main fatty acid (59.5–76.7%). Fatty alcohols range from C16:1 to C24:1, with C20:1 and C22:1 as the primary fatty alcohols, comprising 40.2–45.4% and 45.0–48.3%, respectively. Our data show that the fatty acid and fatty alcohol composition of the wax evaluated in this work was within the reported range and very similar to the previously reported results of the same varieties [6]. Both the commercial batches of Jojoba wax (JD1, JD2, JD3, and JD-4) and the wax obtained from individual Jojoba cultivars (Benzioni, Shiloah, Hatzerim and Sheva) showed high compositional similarity. Previous studies were able to also demonstrate the presence of low levels of the polyunsaturated fatty acids C18:2 and C18:3 (0.1% and 0.23–0.4%, respectively) [37]. However, in the current work, we were unable to detect the presence of these compounds.

Simmondsin is one of the principal active compounds in Jojoba and was suggested to mediate several of its medicinal properties [38,39]. Typically, its level in the wax is low, since it is retained in the remaining cake (Jojoba meal) following the wax extraction process. However, due to its importance and considering its reported medicinal properties, we examined the direct antiviral action of Simmondsin at 0–200 µg/mL. The results presented in Figure 3A,B clearly demonstrate that simmondsin had no noticeable anti-herpes action. Furthermore, we have also evaluated the action of petroleum oil (PO), the main alternative oil used in topical formulations instead of Jojoba. As seen in Figure 3C, PO had a low yet significant impact on HSV-1 plaque formation. In comparison to Jojoba wax, this reduction is low but suggests that the effect may be due to long-chain carbon structures, and the superiority of Jojoba is due to the unique long-chain (C36–C46) esters of fatty acid and fatty alcohol composition, forming a long aliphatic structure. 

To gain more insight regarding the reduced HSV-1 plaque formation in Jojoba-treated groups shown above and to validate the results, two additional tests were carried out: quantification of HSV-1 infection by fluorescent immunohistochemical analysis and by real time PCR (RT-PCR). The upper panel in Figure 4A depicts a representative experiment with anti-HSV-1 antibody (FITC) with Fluorescent DNA Stain (DAPI) counterstaining. The reduction in HSV-1 expression levels upon Jojoba treatment is clearly shown and is comparable to that achieved by acyclovir. Figure 4B shows the quantification of the analysis, further demonstrating Jojoba wax efficacy. Similarly, a decrease in HSV-1 RNA levels was also observed (Figure 4C). In addition, an off-rate experiment was performed; the results presented in Figure 5A show that no inhibitory effect was observed when Vero cells were pre-exposed to the wax and washed out prior to infection, implying that the presence of Jojoba wax is required in the infection step to carry on the protective effect. Lastly, the compatibility of Jojoba wax to be used together with acyclovir was tested. Half of the maximal concentration of both agents were used, as well as a mixture of both. The results in Figure 5B do not support pharmacological compatibility, but rather the dominance of Jojoba in the results. Collectively, these results confirm the reduction in HSV-1 plaque formation shown above. 

Some works have also hypothesized possible bioactivities based on Jojoba’s chemical composition, although these claims were not confirmed by any experimental work [40,41]. From these suggested bioactivities, anti-herpetic activity towards the herpes simplex virus was patented for the Jojoba wax [34] and wax-isolated fatty alcohols [35], and was also reported for Jojoba leaf extract [33]. 

Several potent herbal-based antiviral extracts have been described in traditional medicine. Due to evidence-based meticulous in-vitro, in vivo, and clinical investigation, some gained scientific validation [42,43,44]. These are typically regarded as safer, and with reduced side effects. Here, we demonstrate that Jojoba wax can be used to reduce in vitro HSV-1 infection. Jojoba is commonly used for skin-related conditions, with few side effects and a low skin permeation [14]. The low surface area of application required in HSV-1 infection and the high efficacy demonstrated here also support clinical use. Further studies on the exact molecular pathway and mechanism underlying Jojoba wax effect alongside clinical validation are still required. 

## 3. Materials and Methods

### 3.1. Chemicals and Reagents

Biological Industries (Kibbutz Beit Ha’eMek, Israel) supplied: fetal calf serum (FCS), phosphate-buffered saline (PBS), Penicillin-Streptomycin Solution, Dulbecco’s modified Eagle’s medium (DMEM), and Trypsin–EDTA Solution A. Unless otherwise stated, all reagents were from Sigma-Aldrich Israel. Jojoba waxes and seeds were from Jojoba Desert and Jojoba Valley, Israel. Petroleum oil (PA) was from Johnson and Johnson (Israel). Simmondsin was purchased from BOC Sciences (Shirley, NY, USA).

JD1 through JD4 samples were prepared in Jojoba, Israel in Kibbutz Hatzerim, in routine wax cold press extraction, with an unspecified mixture of seeds from various cultivars. Immediately after the extraction, the wax was put into three 1 L sample glass bottles for further bioactivity assay and chemical composition analyses. Wax from Benzioni, Shiloah, Hatzerim, and Sheva cultivars were similarly prepared.

When indicated, 0.03 g Triton X-100 was added to 0.8 g of Jojoba wax, vortexed, and 170 µL of DDW was added (Jojoba stock solution) to increase its solubility in the cell culture media.

### 3.2. Cell and Microbial Cultures 

Vero cells (African green monkey kidney; CCL-81) and Herpes Simplex Virus type 1 (VR-733) were purchased from the American Type Culture Collection (ATCC; Rockville, MD, USA). The cells were cultured in Dulbecco’s modified Eagle’s medium (DMEM) and supplemented with 5% fetal bovine serum (FBS), 100 μg/mL penicillin, and 100 μg/mL streptomycin, and maintained at 37 °C in a 5% CO_2_ incubator. *Staphylococcus aureus* and *Staphylococcus epidermidis* were purchased from the ATCC (12228 and 33591, respectively) and grown in Luria Broth. *Malassezia furfur* was obtained from the ATCC (14521) and maintained in Brain Heart Infusion (BHI) medium in a shaking bacterial incubator at 32 °C.

### 3.3. Minimum Inhibitory Concentration (MIC) and Anti-Fungal Assay

The streak plate isolation technique was utilized to isolate a single homogenous colony. Then, one colony of each bacterial strain was inoculated from the agar plate into a U-shape falcon tube at a final volume of 4 mL in BHI liquid broth. Cultures were incubated in a bacterial incubator-shaker at 37 °C with shaking at 250 rpm overnight. On the day of the experiment, the cultures were diluted to log phase and transferred to a 96-well plate with or without the compounds (17–850 µg/mL). Kinetic measurements at an optical density of 600 (O.D._600_) were taken at the indicated time points (Tecan infinit 200 pro). Ampicillin at 10 µg/mL was used as a positive control. Similarly, *Malassezia furfur* were diluted to O.D._600_ 0.1 and transferred to a 96-well plate with or without Jojoba wax at 85 and 850 µg/mL. After 24 h, AlamarBlue (Resazurin) was added at a final concentration of 0.004%, and the fluorescence was determined (Excitation 550; Emission 590) as previously reported [45]. Zinc pyrithione (ZPT) at 100 µg/mL was used as a positive control.

### 3.4. Cell Viability Assay

Thiazolyl Blue Tetrazolium Bromide (MTT) powder was dissolved in PBS to generate a 5 mg/mL stock solution, filtered, aliquoted, and kept at −20 °C. Following treatments, the stock solution was diluted at 1:10 in PBS and 150 µL were added to each well. The 96-well cell culture plate was incubated at 37 °C in a 5% CO_2_ incubator for 1 h. Then, the MTT solution was discarded, and an equal volume of 2-propanol was added to elute the dye under general shaking. OD at 570 was measured in a plate reader (Tecan infinit 200 pro).

### 3.5. Plaque Reduction Assay

Vero cells were seeded 150,000 cells/mL in 12-well plates. After 24 h, the cells were infected with virus suspensions to produce approx. 50 plaques per well with or without the indicated treatments. After 1.5 h, the medium was aspirated, and the cells were incubated for 96 h in a complete growth medium. Following incubation, the cells were washed once with PBS and fixed in 1 mL of ice-cold methanol for 10 min. The cells were stained in 0.5% crystal violet (dissolved in 70% methanol) for 30 min and rinsed in DDW. Plaques were counted manually and also presented as PFU/well. Acyclovir at 10 µg/mL was used as a positive control. 

### 3.6. Jojoba Fatty Acid and Fatty Alcohol Profiling 

Jojoba fatty acid and fatty alcohol profiling were performed based on previously reported methods [46,47], with some modifications to avoid high temperatures and prolonged reactions times. Wax samples (50 µL) were added to a 2 mL Eppendorf tube. 150 µL of 3.2% sodium methoxide in methanol (*w/v*) were added and vortexed. Tubes were shaken in a thermoshaker (700 rpm) at 40 °C for 30 min. Next, 100 µL of double-distilled water (DDW) was added, followed by an addition of 1000 µL hexane. The tubes were centrifuged for 2 min at 17,000 G. 800 µL of the supernatant were then transferred to an injection vial supplemented with 200 µL of C17:0 (1 mg/mL, internal standard). 

### 3.7. Chromatographic Conditions

1 µL was injected into an Agilent Technologies GC (model 7890N) equipped with a mass spectrometer detector (model 5977). The carrier gas was helium, at a flow rate of 1 mL/min, on a DB-23 (60 m, 0.25 µm, 0.25 mm) column. The oven temperature was initially 175 °C for 5 min, then increased to 240 °C at 5 °C/min, and held for 9.5 min. Inlet temperature was 250 °C and split ratio 10:1. Fatty acids were identified by comparing retention times with those of standard compounds (FAMEs mix, Supelco, Sigma-Aldrich, Rehovot, Israel). The relative composition of the fatty acids in the waxes was determined as a percentage of total fatty acids.

### 3.8. RNA Isolation and Real-Time PCR

Vero cells were seeded 150,000 cells/mL on 6-well plates. After 24 h, the cells were infected with virus suspensions to produce approximately 50 plaques per well with or without the indicated treatments. After 2 h, the medium was aspirated, and the cells were incubated for an additional 16 h. Then, the cells were washed once with PBS, and total RNA was isolated by RNeasy Mini (Qiagen; Germany), followed by cDNA synthesis using oligo(dT) primers (qPCRBIO, UK). RT-PCR was performed with SYBR green (Sigma-Aldrich Israel), with specific primers (F-CCGAACAACATGGGCCTGAT; R-GGCGGTGCATCCAGTACACAAT) against Glycoprotein D (late gene) of HSV-1 virus, as reported by others [48].

### 3.9. Immunofluorescence Assay

Vero cells were seeded 150,000 cells/mL on coverslips placed in 12-well plates. After 24 h, the cells were infected with virus suspensions to produce approx. 50 plaques per well with or without the indicated treatments. After 2 h, the medium was aspirated, and the cells were incubated for an additional 16 h. Then, the cells were washed once with PBS and fixed with ice-cold methanol for 10 min. Permeabilization was performed with 0.1% Triton X-100, followed by washing with PBS, blocking with 1% bovine serum albumin (BSA) for 20 min, and incubation with FITC-conjugated anti-HSV-1 antibody (ab20437, Abcam, UK) for an additional hour at room tempature. Lastly, a mounting medium with 4,6-diamidino-2-phenylindole (DAPI) was added to the coverslips taken to the fluorescent microscope (Axio Observer Z1; Carl Zeiss, Jena, Germany). Ten fields/treatments were quantified in each group by ImageJ for each repetition.

### 3.10. Statistical Analysis

Results are given as Means ± SEM. Statistical analyses were performed using Student’s *t*-test. *n* = 3; *p* < 0.05 were considered significant. 

## Figures and Tables

**Figure 1 molecules-26-06059-f001:**
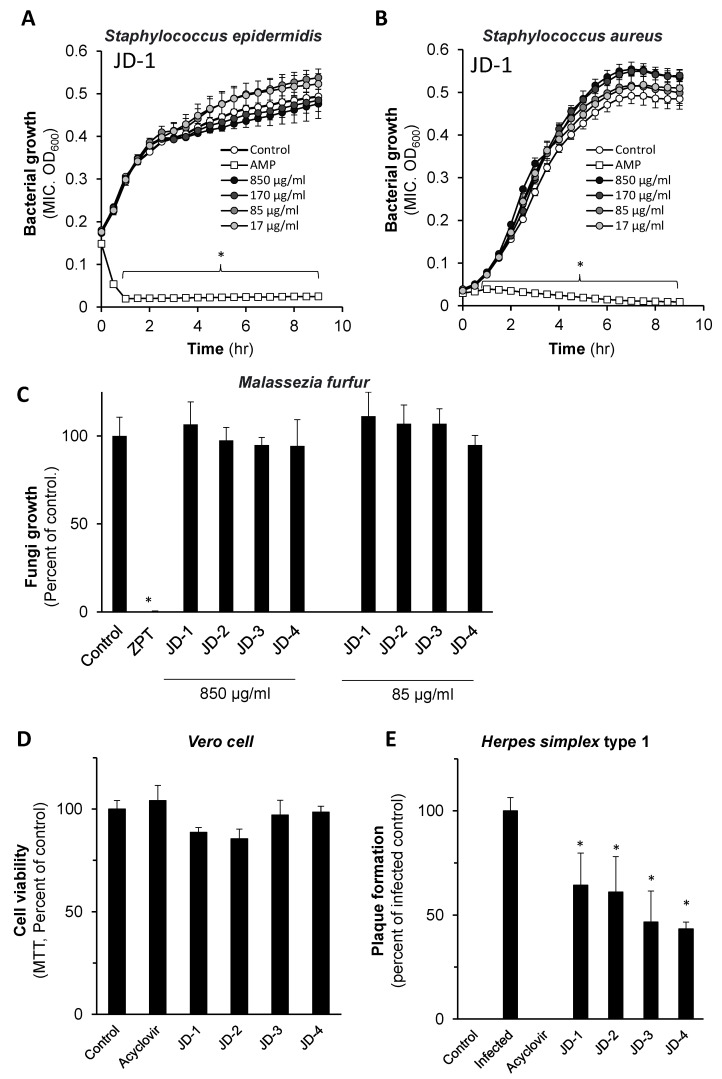
The impact of Jojoba wax on bacterial, fungal, and viral growth. Jojoba wax was added to *S. epidermidis* (**A**), and *S. aureus* (**B**) at the indicated concentrations, and their growth was monitored kinetically with or without the wax supplementation. AMP-ampicillin was used as the positive control (10 µg/mL). The results of representative Jojoba wax (JD-1) are presented. *M. furfur* (**C**) cultures were supplemented with either low (85 µg/mL) or high (850 µg/mL) wax, and fungi growth was assessed by fluorescence of AlamarBlue after 24 h. ZPT- Zinc pyrithione (100 µg/mL) was used as antifungal positive control. Mean ± standard error of the mean (SEM); *n* = 3; * *p* < 0.05 in comparison to the naïve control. (**D**,**E**) present the impact of 850 µg/mL on Vero cell viability or its impact on HSV-1 plaque formation as mentioned in the “materials and methods” section. Mean ± SEM; *n* = 3; *—significant differences, *p* < 0.05 in comparison to the infected control. Acyclovir was used as an antiviral positive control.

**Figure 2 molecules-26-06059-f002:**
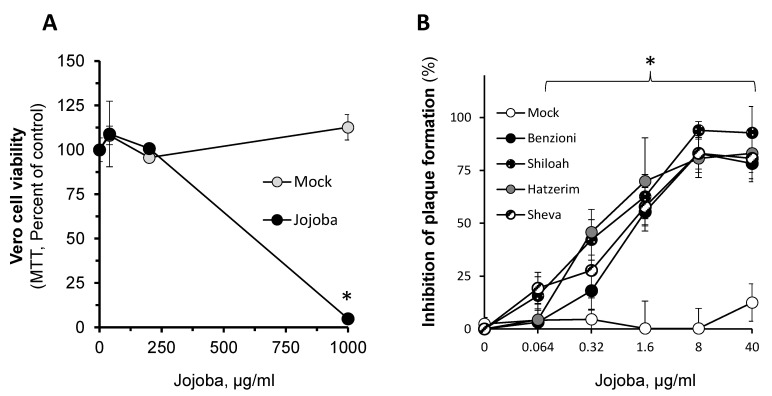
The impact of Jojoba wax on HSV-1 growth. Jojoba wax or vehicle (mock) were added in increasing dilutions to Vero cells. (**A**) Viability was determined by MTT. Mean ± SEM; *n* = 3; * *p* < 0.05 in comparison to the naïve control. (**B**) Dose-response analysis of Jojoba cultivars Benzioni, Shiloah, Hatzerim, and Sheva wax on plaque formation in HSV-1 infected Vero cells was performed as written in the “materials and methods” section. At 40 µg/mL Jojoba, 0.0015% of the surfactant was used. Mean ± SEM; *n* = 3; * *p* < 0.05 in comparison to the infected control.

**Figure 3 molecules-26-06059-f003:**
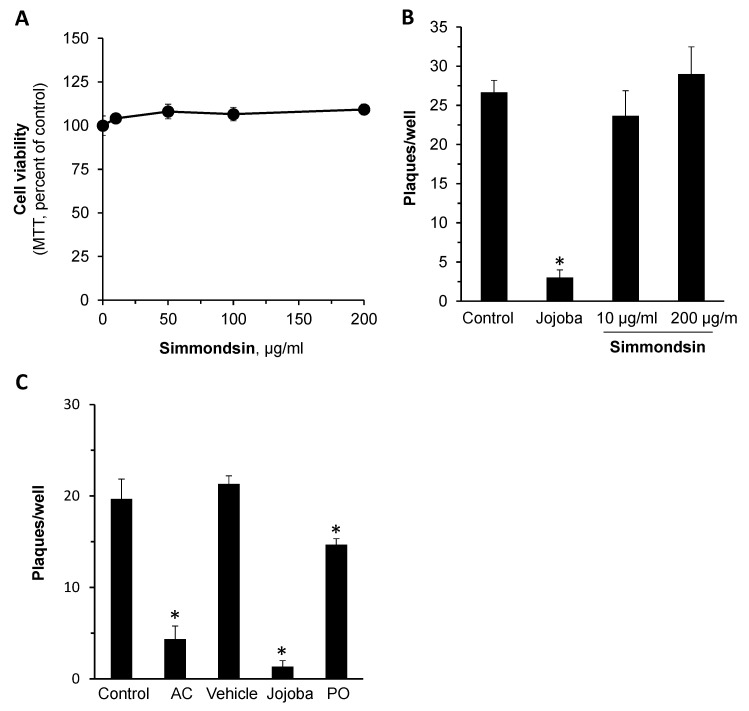
The impact of simmondsin on HSV-1 infection. Vero cells were treated with or without increasing concentrations of Simmondsin for 96 h. Then, cell viability was determined by the MTT assay (**A**). The impact of Simmondsin on HSV-1 infected Vero cells in comparison to Jojoba wax (JD-4; 8 µg/mL) is presented (**B**). A similar test was performed to compare the anti-viral action of Jojoba to 8 µg/mL PA (**C**). Mean ± SEM; *n* = 3; * *p* < 0.05 in comparison to the infected control. AC-acyclovir; PO–petroleum oil.

**Figure 4 molecules-26-06059-f004:**
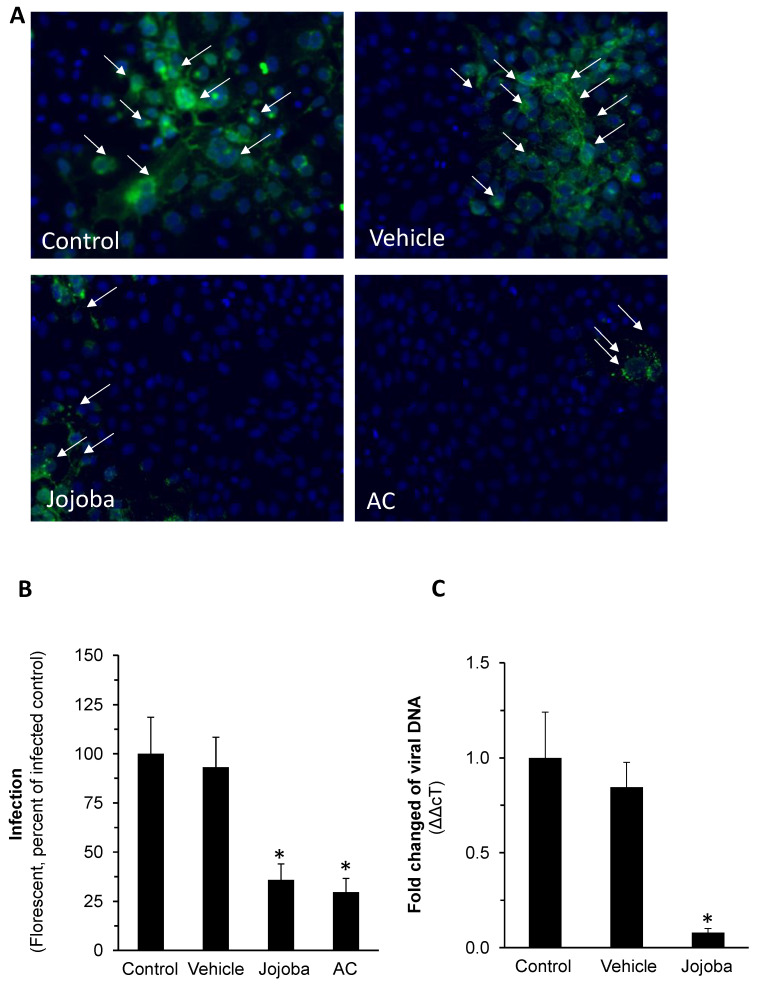
The impact of Jojoba wax on HSV-1 expression level. Infected Vero cells were treated with or without 8 µg/mL of JD-4 wax. After 24 h, the cells were washed, fixed, and an immunocytochemical analysis was performed. Representative images show (**A**) viral protein formation (green) and (**B**) viral protein quantification. DAPI (blue) was used as counterstaining. In addition, RT-PCR was performed after total RNA extraction and cDNA formation (**C**). Mean ± SEM; *n* = 3; * *p* < 0.05 in comparison to the infected control. AC-acyclovir.

**Figure 5 molecules-26-06059-f005:**
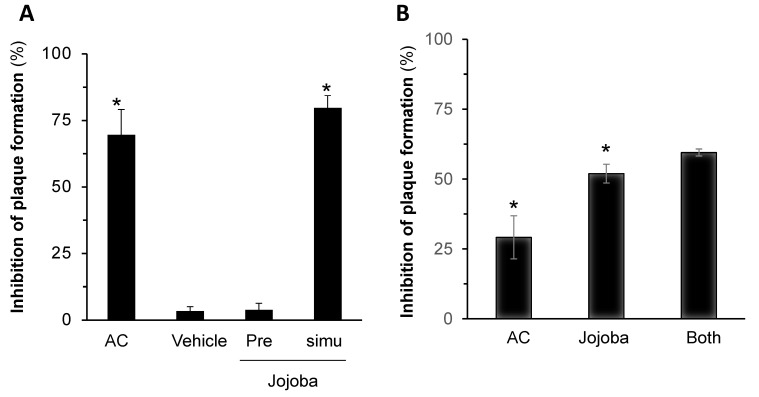
Pharmacological evaluation of Jojoba wax effect on HSV-1 infected cells. Vero cells were treated with or without 8 µg/mL JD-4 before (Pre) or simultaneously (simu) with HSV-1 infection (**A**). After 96 h, plaque formation was quantified as previously described. In addition, the cells were treated with acyclovir (0.25 µg/mL), Jojoba (0.32 µg/mL), or both, and similarly analyzed (**B**). Mean ± SEM; *n* = 3; * *p* < 0.05 in comparison to the infected control.

**Table 1 molecules-26-06059-t001:** Chemical composition of Jojoba wax. Benzioni, Shiloah, Hatzerim and Sheva, as well as commercial batches (JD1–4) used in Figure 1.

	C16:0	C16:1	C18:0	C18:1	C18:2	C20:1	C22:1	C24:1	C18:1 OH	C20:1 OH	C22:1 OH	C24:1 OH
JD1	0.03 ± 0.0	0.15 ± 0.0	0.08 ± 0.0	12.54 ± 0.3	0.46 ± 0.1	71.9 ± 0.3	13.85 ± 0.6	0.98 ± 0.0	0.24 ± 0.0	45.3 ± 0.3	48.7 ± 0.5	5.77 ± 0.2
JD2	0.04 ± 0.0	0.21 ± 0.0	0.06 ± 0.0	13.21 ± 0.1	0.41 ± 0.0	72.5 ± 0.4	12.75 ± 0.6	0.86 ± 0.0	0.21 ± 0.0	50.6 ± 0.1	44.1 ± 0.1	5.04 ± 0.0
JD3	0.06 ± 0.0	0.24 ± 0.0	0.05 ± 0.0	10.73 ± 0.2	0.44 ± 0.1	73.2 ± 0.4	14.15 ± 0.4	1.09 ± 0.1	0.19 ± 0.1	47.2 ± 0.2	47.0 ± 0.6	5.63 ± 0.2
JD4	0.02 ± 0.0	0.37 ± 0.0	0.03 ± 0.0	8.39 ± 0.1	0.26 ± 0.0	79.1 ± 0.1	10.84 ± 0.1	0.94 ± 0.1	0.21 ± 0.1	43.2 ± 0.7	49.0 ± 0.9	7.53 ± 0.1
Benzioni	0.94 ± 0.0	0.26 ± 0.0	0.05 ± 0.0	6.19 ± 0.1	0.02 ± 0.0	76.0 ± 0.4	14.21 ± 0.5	1.78 ± 0.1	0.07 ± 0.0	43.4 ± 0.5	49.6 ± 0.7	6.91 ± 0.0
Shiloah	0.90 ± 0.0	0.22 ± 0.0	0.05 ± 0.0	5.30 ± 0.2	0.11 ± 0.0	76.2 ± 0.1	14.98 ± 0.1	2.18 ± 0.1	0.08 ± 0.0	38.9 ± 0.2	52.3 ± 0.9	8.71 ± 0.2
Hatzerim	1.15 ± 0.0	0.20 ± 0.0	0.06 ± 0.0	7.32 ± 0.3	0.05 ± 0.0	72.8 ± 0.1	16.29 ± 0.4	2.00 ± 0.1	0.12 ± 0.0	44.0 ± 0.3	48.0 ± 0.8	7.09 ± 0.1
Sheva	0.97 ± 0.0	0.25 ± 0.0	0.05 ± 0.0	5.61 ± 0.2	0.00 ± 0.0	78.0 ± 0.3	13.53 ± 0.2	1.55 ± 0.1	0.05 ± 0.0	43.6 ± 0.6	49.7 ± 0.2	6.64 ± 0.0

## Data Availability

Not applicable.

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
