# Peer review of "Anti-Herpes Simplex 1 Activity of Simmondsia chinensis (Jojoba) Wax"

_molecules, 2021, doi:10.3390/molecules26196059_

Round 1

Reviewer 1 Report

The work is interesting, follows the scope of the magazine and can be published. Standard and apparently well-executed techniques were performed. However a major review needs to be carried out:

Article: Anti-herpes Simplex 1 Activity of Simmondsia Chinensis (Jojoba) Wax 
1.Abstract:
- The current research aimed to elucidate the impact of Jojoba wax on skin residential bacterial (Staphylococcus aureus and Staphylococcus epidermidis), fungal (Malassezia furfur), and viral (herpes simplex 1; HSV-1).
Replace viral by virus infection
2. Results and Discussion
- In the sentence according to Al-Ghamdi and colleagues: “extremely high concentration (approx.. 10% of the growth medium) can result in false-positive results due to reduced nutrients, as well as indicative of very low potency compounds with low pharmacological importance” – this hypothesis could be investigated by the authors before being suggested, as it questions the veracity of a work already published. I suggest restricting the comparison on the concentration difference. In this case, could the sources of the wax used in the two studies also influence this difference?
- Cite the concentration tested by Elnimiri and Nimir (2011) on the effect on methicillin-resistant S. aureus.
- The authors report: “yet, to our knowledge, the current study is the first research showing supportive in vitro data for Jojoba wax to those clinical observations” this statement leaves me in doubt whether the research is new and necessary. Authors need to confirm this fact.
- The figures need to identify the tested extracts in the legend. Example, in Figure 1, the caption references only JD-1.
- In Figure 2, substitute Veto cell to Vero cell.
- About the different inhibition results of graphs 1E and 2B. The authors explain that it is due to greater bioavailability with the use of surfactant. If Triton X-100 has detergent action and HSV-1 is an enveloped virus, couldn't the improvement in activity be due to the synergistic action with the surfactant? What concentration of the surfactant when the extract was at 40 µg/ml? Was a control carried out with this concentration of the surfactant?
- Regarding the RT-PCR assay, HSV is a DNA virus. Why did the authors perform the quantification considering the expression and not the reduction of viral DNA? If expression was evaluated, which gene was considered (immediate early, early or late)? Insert the target gene into the methodology.
3. Materials and methods
- The compounds evaluated in the work leave the reader confused. I suggest that the authors further detail the tested extracts: oil, wax, cultivars, Simmondsin. Some of these only appear in the results.
- The acronyms JD and numbers could be modified by the initials of the cultivars, this would make it easier for the reader and would make it more uniform, instead of random numbers (10,11,19,22).
- Acronyms must also be standardized in the text and figures.
- Enter the Vero cell identification reference (CCL-81, E6...)
- Add the concentrations tested in the viability, antibacterial, antifungal and antiviral assays.
- In item 3.6. check the sentences in english
- Standardize ml or mL and µl or µL
- In item 3.9 confirm the term “Permipilization”. I do not know. Also define the acronym RT

Author Response

We thank the reviewer for his/her careful examination of the manuscript. We have made amendments according to the suggestions. (attached)

Reviewer 2 Report

Authors in this manuscript focus on the antiviral activity of the jojoba wax against herpes simplex-1. This manuscript reviews 48 articles and provide a complex survey of current literature dealing with this topic. The topic of this manuscript is up to date, attractive and well suited for Journal - Molecules. The manuscript is well written and divided into three parts. The text is clear and easy to read.  For better understanding authors used five figures and one table. This aid the readers understanding. I suggest checking for some minor spelling mistakes and grammar errors. Otherwise, I  have no essential major regarding this manuscript and recommend it for publication.

Author Response

We thank the reviewer for his/ her effort and positive feedback. We have re-checked the manuscript and (hopefully) amended all grammar and spelling errors)

Reviewer 3 Report

The text bellow contains comments on “Anti-herpes Simplex 1 Activity of Simmondsia Chinensis (Jojoba) Wax”. The manuscript is focused on the elucidation of the impact of Jojoba wax on skin residential bacterial (Staphylococcus aureus and Staphylococcus epidermidis), fungal (Malassezia furfur), and viral (herpes simplex 1; HSV-1).

The manuscript is well written with logically designed experimental parts. The obtained results fully explain the aim of the study.

However, I have some comments listed bellow in case the authors take them into consideration:

Page 2: You should first give the full names of PGE2 and TNFα before to introduce their abbreviations.

Page 3. Reference [31] should be in normal, not in italic style.

Page 3. References [21, 23] should ne before the full stop.

Page 6. In the legend of figure 2 you should remove the yellow coloring of the text.

Why did you investigated the antimicrobial and antiviral effect of simmondsin. It might be principal active component of Jojoba, but I could you did not investigated the presence of this compound in the current study.

Author Response

We thank the reviewer for his/ her positive feedback. We have revised the manuscript according  to the suggestions:

Page 2: You should first give the full names of PGE2 and TNFα before to introduce their abbreviations.

We agree. The full names were added as suggested  

Other important reported bioactivities include a remarkable anti-inflammatory effect via a significant decrease in Prostaglandin E2 (PGE2) content in exudates while preventing Tumor necrosis factor α (TNF-α) formation [17]. Furthermore, Jojoba wax improved acne in a clay face mask formulation [18] and was proved effective in treating acne and psoriasis [19].

Page 3. Reference [31] should be in normal, not in italic style.

We thank the reviewer for observing the incorrect display and corrected it.

Page 3. References [21, 23] should ne before the full stop.

The display was changed, thanks.

Page 6. In the legend of figure 2 you should remove the yellow coloring of the text.

Color was removed, thanks.

Why did you investigated the antimicrobial and antiviral effect of simmondsin. It might be principal active component of Jojoba, but I could you did not investigated the presence of this compound in the current study.

This insightful comment was debated prior to the summation by the research group.  Indeed, simmondsin concentration in the wax was assessed and found to be below the detection level. However, it is a pivotal active compound and we felt that the investigation and presentation (of the negative results) will be of importance to the readers.

Round 2

Reviewer 1 Report

I am satisfied with the authors' corrections/explanations.